# Sol-Gel Synthesized High Entropy Metal Oxides as High-Performance Catalysts for Electrochemical Water Oxidation

**DOI:** 10.3390/molecules27185951

**Published:** 2022-09-13

**Authors:** Muhammad Asim, Akbar Hussain, Safia Khan, Javeria Arshad, Tehmeena Maryum Butt, Amina Hana, Mehwish Munawar, Farhat Saira, Malika Rani, Arshad Mahmood, Naveed Kausar Janjua

**Affiliations:** 1Department of Chemistry, Quaid-i-Azam University, Islamabad 45320, Pakistan; 2National Centre for Physics, Islamabad 44000, Pakistan; 3Department of Physics, The Women University Multan, Multan 66000, Pakistan; 4National Institute of Lasers and Optronics College, Pakistan Institute of Engineering and Applied Sciences, Nilore, Islamabad 45650, Pakistan

**Keywords:** high entropy oxides, sol-gel synthesis, electrocatalysts, catalytic water oxidation, cyclic voltammetry

## Abstract

Hexanary high-entropy oxides (HEOs) were synthesized through the mechanochemical sol-gel method for electrocatalytic water oxidation reaction (WOR). As-synthesized catalysts were subjected to characterization, including X-ray diffraction (XRD), Fourier transforms infrared (FTIR) analysis, and scanning electron microscopy (SEM). All the oxide systems exhibited sharp diffraction peaks in XRD patterns indicating the defined crystal structure. Strong absorption between 400–700 cm^−1^ in FTIR indicated the formation of metal-oxide bonds in all HEO systems. WOR was investigated via cyclic voltammetry using HEOs as electrode platforms, 1M KOH as the basic medium, and 1M methanol (CH_3_OH) as the facilitator. Voltammetric profiles for both equiatomic (EHEOs) and non-equiatomic (NEHEOs) were investigated, and NEHEOs exhibited the maximum current output for WOR. Moreover, methanol addition improved the current profiles, thus leading to the electrode utility in direct methanol fuel cells as a sequential increase in methanol concentration from 1M to 2M enhanced the OER current density from 61.4 to 94.3 mA cm^−2^ using NEHEO. The NEHEOs comprising a greater percentage of Al, ([Al_0.35_(Mg, Fe, Cu, Ni, Co)_0.65_]_3_O_4_) displayed high WOR catalytic performance with the maximum diffusion coefficient, D° (10.90 cm^2^ s^−1^) and heterogeneous rate constant, k° (7.98 cm s^−1^) values. These primary findings from the EC processes for WOR provide the foundation for their applications in high-energy devices. Conclusively, HEOs are proven as novel and efficient catalytic platforms for electrochemical water oxidation.

## 1. Introduction

Water oxidation reaction (WOR), a cleaner and sustainable method for hydrogen production, is a promising substitute for conventional fossil fuels owing to zero or reduced carbon emission, relative simplicity, and its higher energy conversion efficiency [1,2]. The oxidation process is based upon the decomposition of water molecules into oxygen and hydrogen when an electric current passes through it [3,4]. In water electrolysis, hydrogen evolution reaction (HER) occurs at the cathode while oxygen evolution reaction (OER) takes place at the anode in their respective potential windows [5]. The ideal potential difference for the electrochemical water oxidation is 1.23 V between both electrodes [6,7,8]. However, for practical implications, considerably higher voltage is required to achieve a sustainable OER and HER due to the large overpotentials requirements. A multitude of research has been performed to minimize this overpotential by the help of various electrocatalysts. Typically, precious ruthenium and iridium based metal oxides and composites are used for efficient water electrocatalysis however, efforts are being made to develop cost-effective and efficient electrocatalysts for economic water splitting [9,10]. Exploring best performing noble metal-free electrocatalysts is still a huge encounter for satisfactory output.

Recently, transition metal selenides [11,12,13], carbon-fiber-based cobalt nanocatalysts (Co-NC/CF) [14], porous carbon/rGO-composite-based metal-organic frameworks (MOF), and carbon nanotubes (CNT) [15], Co_3_O_4_ nanoparticles-MnO_2_ nanosheets composite [16], 2D transition metal carbide structures [17,18,19], and bimetallic Co-Mo nitride nanosheets have been reported as highly efficient electrocatalysts for boosting up the kinetics of electrochemical oxidation reactions [20,21].

Moreover, high entropy oxides (HEO), an emerging class of active catalytic compounds, have been reported for some photocatalytic and electrocatalytic reactions due to their unique structural and possible tailoring properties [22,23,24]. Positive entropy is the basic pillar behind the formation of stable crystal structures in HEOs [25,26]. High entropy causes the formation of a stable solid solution of a multicomponent system in a single-phase instead of a multiphase mixture of elements [27]. Such materials exhibit high thermal stability, corrosion resistance, and mechanical strength [28]. HEOs are unique because of their multicomponent cation system stabilized by configuration entropies, thereby acquiring a single-phase oxide system. Such formation arises when high sintering temperature increases the entropy of the system that overcomes the positive enthalpy factor and decreases the free energy of formation hence, stabilizing the multi-cationic system in one specific lattice structure [29,30]. The electrochemical response of HEOs depends upon the collective metal-cations matrix constituting the system, thus providing a tailored platform to optimally apply the electrochemical properties just by exchanging/adding any elemental composition based on the entropy-driven stabilization. HEOs can be formulated with transition metals (TM), rare earth metals (REM), or both TMs and REMs [31]. Additionally, HEOs can be equiatomic (EHEOs) or non-equiatomic (NEHEOs) based upon the atomic ratios of different elements constituting the metal oxide lattice. These compounds exhibit commendable room temperature Li-ion conductivities, narrow band gaps, and colossal dielectric constants [32,33]. The rapid evolution of this field of HEOs is also governed by the availability of different possible synthetic routes, i.e., co-precipitation, auto-combustion sol-gel method, hydrothermal routes, etc. [34,35]. Novel HEO (constituting Mn, Cr, Co, Ni, and Fe) perovskites (both equiatomic and non-equiatomic) have been reported recently on the catalytic applications of HEOs, including CH_4_ oxidation, alcohol oxidation, desulfurization, CO_2_ reduction, and water oxidation [36]. A five-element-based quinary np-Al, Ni, Co, Ir, and Mo HEO has been envisioned as efficient water splitting electrocatalyst [37]. Furthermore, a novel (Be, Mg, Ca, Sr, Zn, Ni)_3_O_4_ hexanary HEO has been developed for high performance water electrolysis [38]. However, a few studies have been published so far on HEOs for water oxidation that revealed the super high activity of these compounds towards electrocatalytic water oxidation.

The foundation principle of HEOs’ formation lies in the entropy of mixing or stabilization, which is termed configurational entropy and is denoted by ΔS_config_. Positive ΔS_config_ is the vital tool responsible for the formation of a single phase of as-synthesized hexanary HEOs [39]. Estimation of entropy can be calculated by mole fractions of each element present in the oxide system both at cationic and anionic sites by applying Equation (1) [38].
(1)$$\Delta S\left(\mathrm{config}\right)=\sum_{i=1}^{\mathrm{Ni}}x_{i}{\;\mathrm{lnx}}_{i}+\sum_{j=1}^{\mathrm{Nj}}x_{i}{\;\mathrm{lnx}}_{j}$$

Here, x_i_ and x_j_ are the mole fractions at cationic and anionic sites. x_i_ is estimated from the sum of mole fractions of all the six elements present in each oxide system. x_j_ is mole fraction of oxygen/oxide ion and is taken as 1, where (ln 1 = 0); thus, the anionic site exerts a negligible effect on the entropy of mixing [40]. Hence, the whole entropy of the system depends upon the cationic site comprising metallic elements only. In the present study, the configurational entropy calculated for equiatomic and non-equiatomic HEOs are found to be 1.8R and 1.7R, respectively. The materials with ΔS_config_ ≥ 1.5R are classified as “high entropy materials”, 1.5R > ΔS_config_ ≥ 1R are classified as “medium entropy materials”, and the materials having ΔS_config_ < 1R are termed as “low entropy materials” [41]. Therefore, values of entropy (in the 1.8R and 1.7R range) found for as-synthesized hexanary oxides indicated the formation of high entropy materials (HEMs). Moreover, the ΔS_config_ value is maximum when the HEO is equiatomic, and it decreases when elemental mole fractions are disturbed. Furthermore, all equiatomic HEOs have same entropy value and all non-equiatomic HEOs have same ΔS_config_ as it merely depends on the number of the constituents rather than any other chemical effect or nature of the constituents.

Also, only a few reports have been published for HEOs in electrochemical (EC) and energy applications [42]. Herein, we report the sol-gel synthesis of different series of multicomponent-based HEOs (a) equiatomic HEOs (EHEOs) and (b) non-equiatomic HEOs (NEHEOs) and their potential catalytic usage for water electro-oxidation (WOR) via cyclic voltammetric (CV) technique. All HEOs containing the equiatomic elemental and non-equiatomic metal compositions were synthesized by the aqueous sol-gel method using citric acid as the chelating agent along with ammonia solution to maintain the pH. Oxides were further subjected to characterization using XRD and FTIR analysis. Subsequently, the catalytic performance of as-synthesized HEOs was investigated for electrochemical water oxidation (WOR) in basic media in the presence of methanol via cyclic voltammetry. Resultantly, the as-synthesized HEOs have been observed as efficient water oxidation electrocatalysts.

## 2. Materials and Methods

### 2.1. Chemicals

Aluminum nitrate, (Al(NO_3_)_3_·9H_2_O) (99.99%), magnesium nitrate, (Mg(NO_3_)_2_·6H_2_O) (99%), iron nitrate, (Fe(NO_3_)_3_·9H_2_O) (98%), copper nitrate, (Cu(NO_3_)_2_·3H_2_O) (99%), nickel nitrate, (Ni(NO_3_)_2_·6H_2_O) (99.99%), cobalt nitrate, (Co(NO_3_)_2_·6H_2_O) (98%), ammonium nitrate, (NH_4_NO_3_) (99%), cadmium nitrate, (Cd(NO_3_)_2_·4H_2_O) (98%), chromium nitrate, (Cr(NO_3_)_3_·9H_2_O) (99%), manganese acetate, (Mn(CH_3_COO)_2_) (98%), zinc nitrate, (Zn(NO_3_)_2_·9H_2_O) (98%), citric acid, (C₆H₈O₇) (99%), acetone, (CH_3_OCH_3_) (99.5%), ammonium hydroxide, (NH_4_OH) (25% NH_3_), methanol, (CH_3_OH) (99.99%) and ethanol, (C_2_H_5_OH) (99.99%) were used in synthesis and experimental work process. All the chemicals were of Sigma Aldrich AR grade and were utilized as received without further purification. All the experiments were performed with freshly prepared solutions in deionized water (DI).

### 2.2. Synthesis of HEOs

Both the EHEOs and NEHEOs were synthesized by the sol-gel auto-combustion method using citric acid as the complexing/gelling agent and ammonia solution to maintain pH at 7 as previously reported [35,42]. In order to prepare an equiatomic HEO, 10.4 mmol solutions of each metal precursor solution were prepared separately in 10 mL DI water and were mixed to attain homogenous solution. For non-equiatomic oxides, 12.5 mmol solution of one component is added to the mixture while keeping the major component at constant composition. Mixture was kept stirring at 90 °C for 12 h. until a viscous homogenous gel was formed. The gel was further completely dried on the hot plate at 80 °C, followed by calcination at 800 °C for 9 h for complete removal of nitrates and other tangling ingredients. The powder obtained was pulverized in agate mortar, and finally, a fine powder was achieved.

### 2.3. Instrumentation

XRD was performed to examine the average particle size, crystal shape, and phase study of as-synthesized HEOs using PANalytical diffractometer equipped with Cu Kα radiation having λ of 1.54 Å, working in 10°–80° range of 2θ and setting the step size of 0.02°. FTIR spectra for all the prepared HEOs were obtained in the wavelength range between 400–4000 cm^−1^ using Nicolet 5PC instrument, where the samples were mixed with KBr powder. Electrochemical measurements were accomplished using Gamry-potentiostat interface-1000. Three electrode system was set for cyclic voltammetric measurements. Glassy carbon electrode (GCE) was used as working electrode, platinum (Pt) wire as counter electrode and silver/silver chloride (Ag/AgCl 3M KCl) was employed as reference electrode. GCE was modified with HEO powder bound by Nafion, which acts as polymer electrolyte membrane (PEM) for electron exchange and also as a binder.

### 2.4. Electrode Modification

GCE was fabricated with weighed amount of catalyst powder for voltammetric studies using drop-cast method [43]. Prior to electrode modification, GCE was first cleaned with alumina slurry and then rinsed with ethanol. A total of 5 mg of finely ground catalyst powder was poured on GCE shining surface, followed by drop-casting of 5% Nafion solution. The fabricated electrode was kept for drying in oven at 55 °C for 30 min before EC application.

## 3. Results and Discussion

### 3.1. X-ray Diffraction

Powdered XRD patterns were recorded for all the as-synthesized HEOs, and the diffraction patterns are shown in Figure 1. It can be visualized that the structural properties are influenced by the incorporation of any new individual element in the oxide system. It is evident that the increase in any element’s atomic composition affects the peak intensity. Herein, the weight percentages of Cu, Ni, and Co are kept constant, while the increase in Fe, Mg, and Al has been investigated one by one. Figure 1 (i) corresponds to the respective equiatomic HEO system, while Figure 1 (ii–iv) expresses the increase in Fe, Mg, and Al, respectively. The enhancement in peak intensity at 36° in Figure 1 (ii) is indexed to iron and refers to an increase in iron content [44]. A decrease in peak intensity at 43° and 62° is observed while increasing the Mg content as presented by the XRD pattern in Figure 1 (iii) [45]. Peak intensities increased at 31° and 38° in Figure 1 (iv), corresponding to an increase in the Al oxide atomic ratio in the HEO lattice [46,47]. All major peaks are indexed to hkl values and correspond to the crystallinity of HEOs. Similar XRD analyses were also performed for all NEHEO compositions.

Furthermore, the average crystallite sizes (D_av_) of all the as-synthesized HEOs were estimated using XRD data and analysis from all the reflection peaks. Widths of XRD reflection patterns relate to average crystallite size [48]. D_av_ is calculated using the Debye–Scherrer formula expressed in Equation (2) [49].
(2)$$D_{\mathrm{av}}\;\left(\mathrm{nm}\right)=\frac{57.2\;k\;\lambda }{\beta \;\mathrm{Cos}\;\theta }$$

Here, *k* represents the constant (0.94), λ corresponds to the wavelength of radiation (for Cu-Kα, λ = 0.1541 nm), *β* denotes the full-width-half-maximum (FWHM) of the diffraction peaks, and θ relates to the Bragg’s diffraction angle. The estimated D_av_ values for all the as-synthesized equiatomic and non-equiatomic HEOs are enlisted in Table 1. Non-equiatomic HEOs exhibited smaller D_av_ than the equiatomic compositions signifying that the former might give higher electrocatalytic responses. From serial No. 3 to 5 and 6 to 8 in Table 1, it is perceived that the HEO system with higher Al contents in respective series possessed the smallest average crystallite size value, thus, indicating that the higher Al molar ratio will produce a more active HEO catalyst comparative to its relative compositions, this attributes to the lower ionic radius of the Al^3+^ ion which may cause the lattice size shrinkage and lower of the cell parameters [27]. Moreover, NEHEOs (serial No. 9 to 12 in Table 1) represent the comparison of similar oxide compositions just by increasing the molar ratio of Cu and Mn from 0.25 to 0.35, decreasing the average crystallite size of NEHEOs. It is merely indicating that the oxides with higher Cu and Mn contents compared to their other compositions should provide higher catalytic output towards WOR.

### 3.2. FTIR Analysis

FTIR analysis is performed as a qualitative tool for the investigation of functional groups. The FTIR spectra for all the as-synthesized EHEOs and NEHEOs were recorded in wavelengths ranging from 450 to 4000 cm^−1^ on Nicolet Analytical Instrument. FTIR comparative analysis of EHEOs and NEHEOs is shown in Figure 2a. Absorption peaks between 400–700 cm^−1^ indicate the formation of metal oxide bonds. Moreover, it can be observed that the FTIR peak position slightly shifted by changing the elemental compositions of the HEOs. Replacing the elements in the oxide systems, i.e., EHEOs, exhibited pronounced variations in peaks as compared to just increasing the atomic ratios of oxides within the series, i.e., NEHEOs. Figure 2b represents the comparative FTIR spectra of unsintered and sintered (completely dried) HEO gel. All unsintered spectra depicted (Figure 2b (ii)) the characteristic peaks at 1302 cm^−1^ and 1416 cm^−1^, which correspond to vibrational modes of nitrate (NO_3_^–^) and acetate (COO^–^) ions [50,51]. It is evident that finely dried sintered oxide shown in Figure 2b (i) did not absorb around 3100 cm^−1^ and 1400 cm^−1^, indicating the successful removal of moisture and acetates [52].

### 3.3. SEM Analysis

SEM micrographs for NEHEOs are presented in Figure 3, and these images revealed the spongy/agglomerated structure of HEO nanoparticles, which may be attributed to high-temperature sintering conditioning [38].

### 3.4. Electrochemical Studies

#### 3.4.1. Cyclic Voltammetric Responses of Equiatomic HEOs

Electrochemical studies were carried out to evaluate the electrocatalytic properties of hexanary HEOs through cyclic voltammetry (CV), which is the most fundamental technique in electrochemistry that offers a prompt assessment of the redox processes [53], their potential, and the influence of electrolyte on the respective chemical processes. Variation in peak current intensity with changes in the reaction system, scan rate, and electrode materials has been investigated. Figure 4a shows the voltammetric profile using (Fe, Al, Mg, Cd, Cr, Mn)_3_O_4_ EHEO in 1M KOH in the presence and absence of methanol. Methanol has been used here as the facilitating agent that enhanced the charge transfer across the electrode and reduced the onset potential hence the oxidation peak current. Figure 4b displays the dependence of current upon scan rate using different EHEO comprising different elemental compositions. It is observed that current increases with an increase in scan rate where the maximum current output was delivered by (Fe, Al, Mg, Cd, Cr, Mn)_3_O_4_ when EHEOs are taken into consideration.

#### 3.4.2. Electrochemical Responses of NHEOs

Non-equiatomic are the oxides where one element possesses 35% of the whole oxide system while the other 65% composition is comprised of five different elements with equal weightage. In Figure 5a–c, peak current dependence upon scan rate has been shown for Cd-, Cr-, and Mn-based compositions, i.e., Fe_35_[(Al, Mg, Cd, Cr, Mn)_65_]_3_O_4_, Mg_35_[(Al, Fe, Cd, Cr, Mn)_65_]_3_O_4_ and Al_35_[(Fe, Mg, Cd, Cr, Mn)_65_]_3_O_4_, respectively. It has been observed that the non-equiatomic HEO with higher aluminum oxide content (Figure 5c) displayed the maximum WOR current compared to other compositions in the series. Moreover, the sensitivity of current with changing scan rate can also be observed through linear sweep voltammograms shown in Figure 5d. Comparative CV analysis for Cd-, Cr-, and Mn-based HEOs has been shown in Figure 5e, which clearly refers to Al_35_[(Fe, Mg, Cd, Cr, Mn)_65_]_3_O_4_ as the best catalyst in its series. Similar observations are made for Cu-, Ni-, and Co-based non-equiatomic HEO compositions as given in Figure 5f, which also shows that the HEO with higher aluminum oxide content gave the highest catalytic response towards water oxidation. The catalytic output can be associated with the average crystallite size, D_av_, from the XRD analysis. The compositions with higher Al content possess the smallest D_av_ in their respective series, which may correspond to its higher catalytic efficiency in WOR.

#### 3.4.3. Effect of Increase in Element’s Weight Percentage (25–35%)

High entropy oxides have been synthesized in another compositional manner to study the influence of change in one element in the hexanary oxide while keeping the other elements constant. Herein, the weight percentage of Cu was increased from 25–35%, and the weightage of the other five elements decreased from 75–65%. A similar observation was also made for increasing the percentage of Mn in the hexanary HEO. An increase in anodic current with changing scan rate using Cu_35_(Al, Fe, Co, Cr, Ni)_65_]_3_O_4_ can be observed in Figure 6a. A similar trend in voltammetric profile was also observed for the other three compositions in series, i.e., Cu (25%), Mn (35%), and Mn (25%) in (Al, Fe, Co, Cr, Ni)]_3_O_4_. Comparative voltammograms for this series have been presented in Figure 6b, which explains the behavior of electrode materials. The maximum peak current corresponds to Cu (35%) and Mn (35%) NEHEOs; however, the minimum overpotential is displayed by Mn (35%) NEHEO. Therefore, Mn (35%) can be considered the best composition of NEHEO for water oxidation electrocatalysis. It is also perceived that an increase in Mn and Cu percentage in a constant HEO system would result in enhanced catalytic responses.

#### 3.4.4. Effect of Methanol Concentration on WOR

Methanol is used as a facilitator in WOR catalysis as the peak current was pronouncedly improved, and onset potential was decreased in the presence of methanol in the basic medium [43]. Normally, the scan rate studies were performed in 1M CH_3_OH; however, the impact of the methanol concentration range was also investigated. Remarkably, increasing the methanol concentration caused a significant increase in anodic peak current. The voltammetric profiles obtained from variation in methanol concentration using both equiatomic and non-equiatomic HEOs can be visualized in Figure 7a,b, respectively. A sequential increase in methanol concentration from 1 to 2M enhanced the OER current from 4.3 to 6.6 mA using NEHEO, as shown in Figure 7b. The non-equiatomic HEOs are proved as more efficient electrocatalysts. In this way, HEOs can be optimally used as performing catalysts for electrochemical water oxidation in the presence of high concentrations of methanol.

#### 3.4.5. Kinetics of Water Oxidation over HEOs

A kinetic analysis can be made based on peak current variation with different parameters such as the scan rate, concentration of electrolyte, or facilitating agent. Here, diffusion coefficient and apparent rate constant values have been estimated from cyclic voltammetry using Randles–Sevcik equation (Equation (3)) [43] and Reinmuth equation (Equation (4)) [54], respectively.
I_p_ = (2.99 × 10^5^) n {(1 − α) n_α_} ^1/2^ A D° ^1/2^ C ʋ^1/2^(3)
(4)$$I_{P}=0.227\;n\;F\;A\;C\;k{}^{\circ}$$
whereas I_p_ represents the anodic current, n is the number of electrons evolved in the oxidation process (4 in water oxidation), A represents the active surface area of the working GCE (0.07 cm^2^), D° denotes the diffusion coefficient in cm^2^ s^−1^, C stands for concentration of methanol in molcm^−3^, ν indicates the scan rate in mV s^−1^, F corresponds to Faraday’s constant (96,485 C mol^−1^) and k° represents the heterogeneous rate constant.in cms^−1^.

The diffusion coefficient, D° was estimated from the slope of the Randles–Sevcik plot in Figure 8a, while the heterogeneous rate constant, k° was retrieved from the slope of the Reinmuth plot presented in Figure 8b. Both these graphs are associated with {Fe_0.35_ (Al, Mg, Cd, Cr, Mn)_0.65_}_3_O_4_ NEHEO composition, while D° and k° were estimated for all the synthesized HEO-based systems using the Randles–Sevcik and Reinmuth plots. The retrieved values for D° and k° are enlisted in Table 2.

#### 3.4.6. The Normalized Current Density

The normalized current density (J in mA cm^−2^) is an important performance parameter to measure and compare the intrinsic catalytic activity of the envisioned materials. To have a common comparison balance, the peak current and Ipa values, as retrieved from the CVs recorded at 100 mV s^−1^, were normalized with the geometric electrode area and symbolized as Jpa in mA cm^−2^ units. This parametric comparison clearly indicates a correspondence with the particle sizes and compositions that the smaller the particle size, the better suited the material for electrocatalysis [55]; see Table 3. In all, NEHEOs speciated in better WOR performance than the EHEOs, apparently owing to the availability of better catalytic surfaces.

It has been reported that the water splitting as OER activity posed only a maximum of ~40 mA cm^−2^ current density using novel and state-of-the-art catalysts; IrW NDs, IrW/C, Ir/C, Pt/C, etc. [55]. Ali et al. reported formic acid catalysis using high-class materials comprising Pd composited with reduced graphene oxide [15]. The formic acid anodic peak was achieved up to 0.8 mA in H_2_SO_4_ at a scan rate of 20 mV s^−1^; in contrast to these studies, the OER peak current of up to 4–6 mA was achieved using HEOs. It also infers that the HEOs can be used for further EC processes, including methanol oxidation.

Compared to the above-mentioned ideal catalysts, the envisioned HEOs can lead to high performance as the obtained Jpa values are 60 mA cm^−2^ with a distinct WOR peak in the CV profiles.

#### 3.4.7. Stability Test

The stability test for the fabricated electrodes was conducted in 1M KOH, applying 1.4 V as step-and delay potential for 3600 s in a chronoamperometric way, presented in Figure 9 for AS-A and AK-A NEHEOs. The catalytic platforms were stable over the experimental time scale and would be useable over a period of 1–2 days with the same efficiency.

## 4. Conclusions

High entropy oxides comprising six different elements were synthesized using conventional and facile sol-gel routes involving nitrates of all the elements. These HEOs displayed sharp peaks in XRD analysis, thus indicating the fine crystalline structure, while significant infrared absorbance in the 500 to 700 cm^−1^ range in FTIR spectral analysis referred to the successful formation of metal-oxide bonds. Herein, the first report on the utility of hexanery HEOs as anode materials for water oxidation is being presented, and their electrocatalytic performance was investigated by cyclic voltammetry. All the systems were comparatively conductive and facilitated the EC process for water oxidation, with the observation that non-equiatomic HEOs displayed higher catalytic output current than the equiatomic HEOs. Furthermore, the compositions with higher Cu and Al contents exhibited greater efficiencies toward catalysis. Herein, [Al_0.35_(Mg, Fe, Cu, Ni, Co)_0.65_]_3_O_4_ is observed as the best performer NEHEO owing to higher peak current response, high diffusion coefficient, and rate constant values for WOR. Subsequently, owing to a simple synthesis scheme, cost-effectiveness, and easily accessible elements, the hexanary high entropy oxides could be used as super-efficient and high-performance catalysts for electrochemical water oxidation.

## Figures and Tables

**Figure 1 molecules-27-05951-f001:**
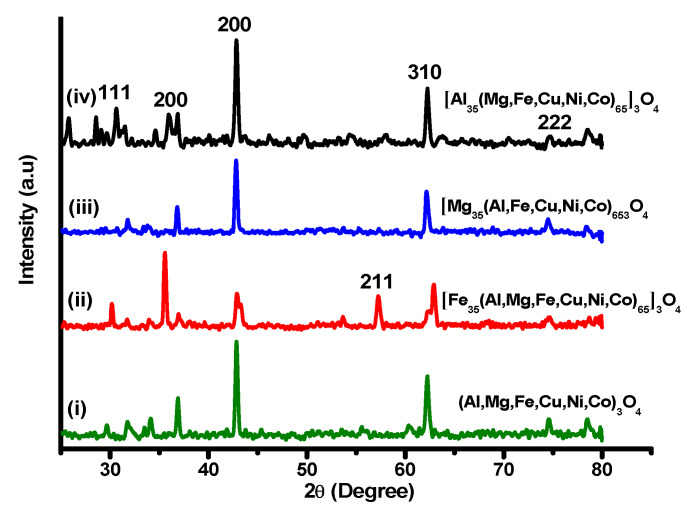
XRD patterns for equiatomic HEOs (i) and non-equiatomic HEOs (ii–iv).

**Figure 2 molecules-27-05951-f002:**
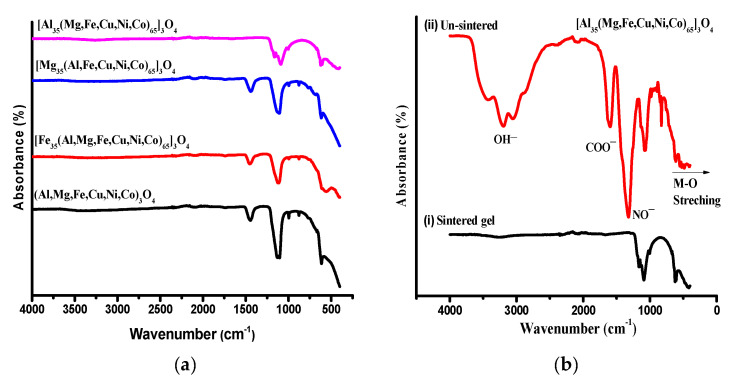
FTIR spectra of EHEOs (**a**), sintered (i) and un-sintered. (ii) [Al_35_(Mg, Fe, Cu, Ni, Co)_65_]_3_O_4_ NEHEO (**b**).

**Figure 3 molecules-27-05951-f003:**
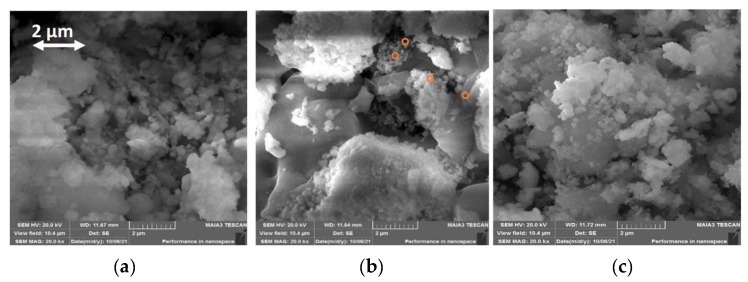
SEM images of NEHEOs [Al_0.35_(Mg, Fe, Cd, Cr, Mn)_0.65_]_3_O_4_ (**a**) [Al_35_(Mg, Fe, Cu, Ni, Co)_65_]_3_O_4_ (**b**); the orange circles signify the nanosized spheroids) [Cu_0.35_(Al, Fe, Co, Cr, Ni)_0.65_]_3_O_4_ (**c**) at 2 µm resolution.

**Figure 4 molecules-27-05951-f004:**
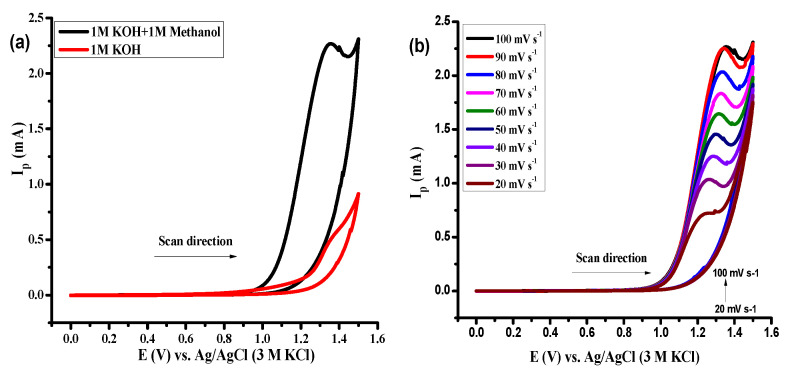
Cyclic voltammograms for equiatomic HEOs; (Fe, Al, Mg, Cd, Cr, Mn)_3_O_4_ in 1M KOH with and without methanol (**a**) and (Fe, Al, Mg, Cd, Cr, Mn)_3_O_4_ at different scan rates from 20–100 mV s^−1^ (**b**).

**Figure 5 molecules-27-05951-f005:**
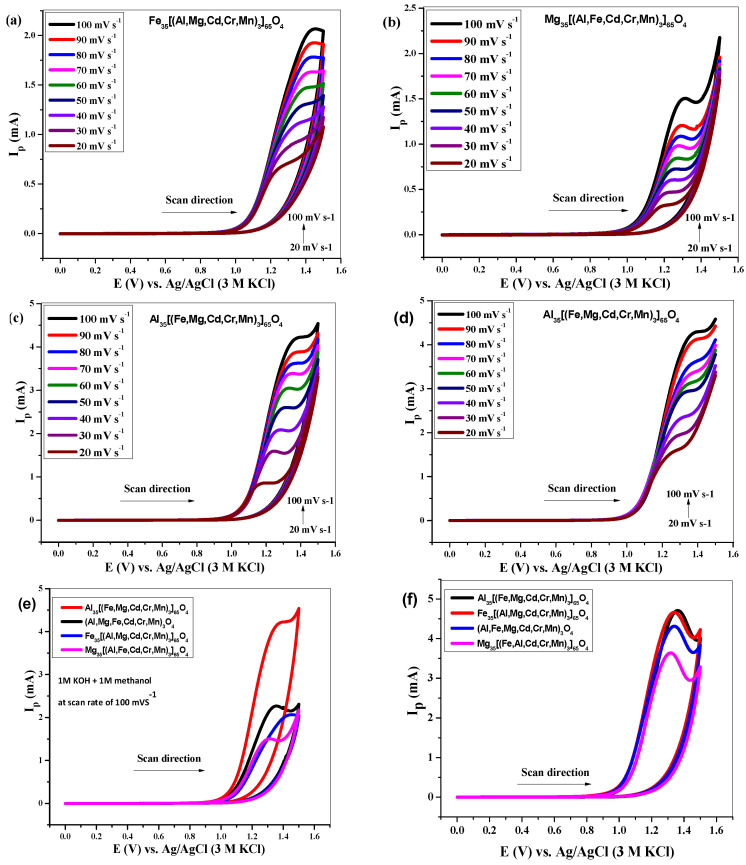
Cyclic voltammetry using [Fe_35_(Al, Mg, Cd, Cr, Mn)_3_)_65_]O_4_ (**a**), [Mg_35_(Al, Fe, Cd, Cr, Mn)_3_)_65_]O_4_ (**b**), [Al_35_(Fe, Mg, Cd, Cr, Mn)_3_)_65_]O_4_ (**c**), LSV for [Al_35_(Fe, Mg, Cd, Cr, Mn)_3_)_65_]O_4_ (**d**), comparison of [(Al, Fe, Mg, Cd, Cr, Mn)_x_]_3_O_4_ (**e**) and comparison of [(Al, Fe, Mg, Cu, Ni, Co)_x_]_3_O_4_ (**f**) HEOs in 1M KOH + 1M CH_3_OH.

**Figure 6 molecules-27-05951-f006:**
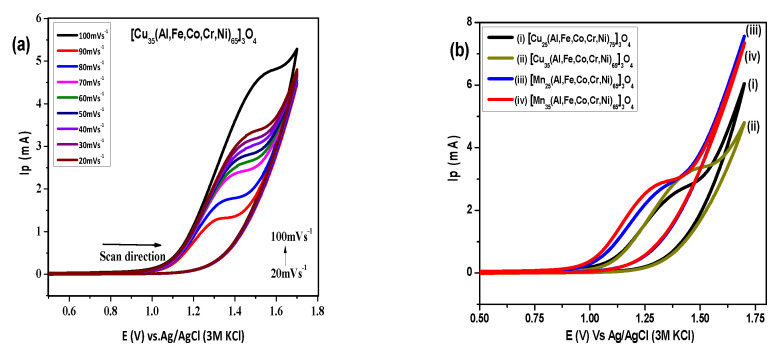
Cyclic voltammograms of water oxidation in 1M KOH + 1M CH_3_OH using [Cu_35_(Al,Fe,Co,Cr,Ni)_65_]_3_O_4_ (**a**), and comparative cyclic voltammograms for different percentage compositions of Cu and Mn in (Al,Fe,Co,Cr,Ni) HEO (**b**).

**Figure 7 molecules-27-05951-f007:**
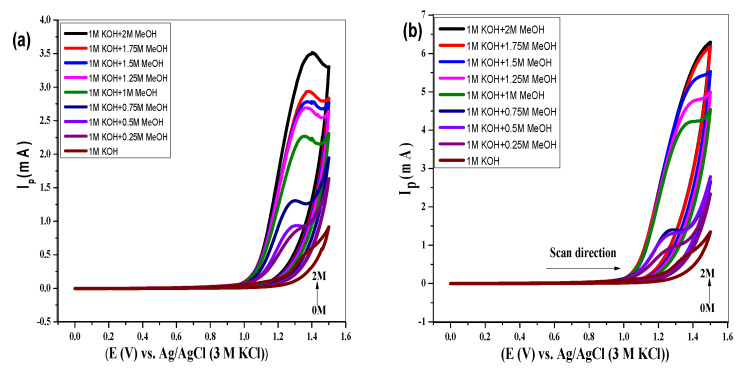
Cyclic voltammograms using equiatomic (Al, Fe, Mg, Cd, Cr, Mn)_3_O_4_ (**a**) and non-equiatomic [Al_35_(Fe, Mg, Cd, Cr, Mn)_65_]_3_O_4_ electrodes (**b**) in 1M KOH at 100 mV s^−1^ showing variation in peak current with increasing the methanol concentration.

**Figure 8 molecules-27-05951-f008:**
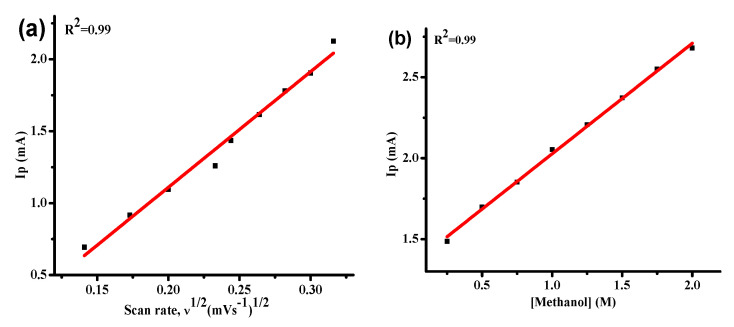
Randles–Sevcik plot (**a**) and Reinmuth plot (**b**) for {Fe_0.35_(Al, Mg, Cd, Cr, Mn)_0.65_}_3_O_4_ NEHEO.

**Figure 9 molecules-27-05951-f009:**
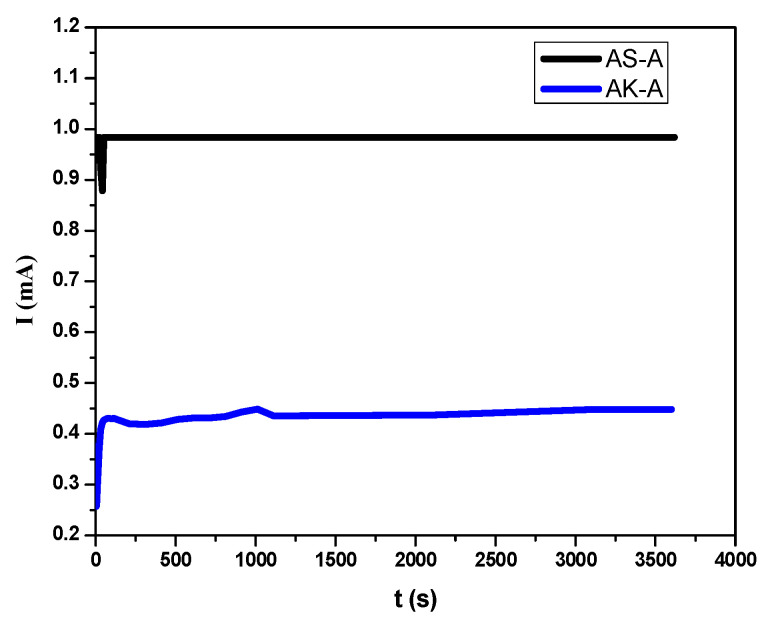
Chronoamperometric stability tests for AS-A and AK-A NEHEOs in 1M KOH for 3600 s.

**Table 1 molecules-27-05951-t001:** Calculated average crystallite sizes of HEOs using Debye–Scherer equation.

Serial No.	Catalyst/(Codes)	D_av_ (nm)
Equiatomic HEOs		
1	(Fe, Al, Mg, Cd, Cr, Mn)_3_O_4_/(AK-1)	20.4
2	(Fe, Al, Mg, Cu, Ni, Co)_3_O_4_/(AS-1)	24.4
Non-Equiatomic HEOs		
3	[Mg_35_(Fe, Al, Cd, Cr, Mn)_65_]_3_O_4_/(AK-M)	17.2
4	[Fe_35_(Al, Mg, Cd, Cr, Mn)_65_]_3_O_4_/(AK-F)	19.3
5	[Al_35_(Fe, Mg, Cd, Cr, Mn)_65_]_3_O_4_/(AK-A)	12.0
6	[Mg_0.35_(Al, Fe, Cu, Ni, Co)_0.65_]_3_O_4_/(AS-M)	30.2
7	[Fe_0.35_(Al, Mg, Cu, Ni, Co)_0.65_]_3_O_4_/(AS-F)	25.7
8	[Al_0.35_(Mg, Fe, Cu, Ni, Co)_0.65_]_3_O_4_/(AS-A)	23.3
9	[Cu_0.35_(Al, Fe, Co, Cr, Ni)_0.65_]_3_O_4_ /(MMCu-35)	4.85
10	[Cu_0.25_(Al, Fe, Co, Cr, Ni)_0.75_]_3_O_4_ /(MMCu-25)	5.16
11	[Mn_0.35_(Al, Fe, Co, Cr, Ni)_0.65_]_3_O_4_ /(MMMn-35)	3.36
12	[Mn_0.25_(Al, Fe, Co, Cr, Ni)_0.75_]_3_O_4_ /(MMMn-25)	7.14

**Table 2 molecules-27-05951-t002:** Diffusion coefficient and rate constant values of water oxidation reaction using HEOs catalysts.

Serial No.	Catalyst	D° × 10^−8^ (cm^2^ s^−1^)	k° × 10^−4^ (cm s^−1^)
Equiatomic HEOs			
1	(Fe, Al, Mg, Cd, Cr, Mn)_3_O_4_	2.74	3.04
2	(Fe, Al, Mg, Cu, Ni, Co)_3_O_4_	9.31	7.55
Non-Equiatomic HEOs			
3	[Mg_35_(Fe, l, Cd, Cr, Mn)_65_]_3_O_4_	1.15	1.89
4	[Fe_35_(Al, Mg, Cd, Cr, Mn)_65_]_3_O_4_	2.47	2.66
5	[Al_35_(Fe, Mg, Cd, Cr, Mn)_65_]_3_O_4_	9.29	5.44
6	[Mg_0.35_(Al, Fe, Cu, Ni, Co)_0.65_]_3_O_4_	4.87	5.66
7	[Fe_0.35_(Al, Mg, Cu, Ni, Co)_0.65_]_3_O_4_	9.72	7.59
8	[Al_0.35_(Mg, Fe, Cu, Ni, Co)_0.65_]_3_O_4_	10.90	7.98
9	[Cu_0.35_(Al, Fe, Co, Cr, Ni)_0.65_]_3_O_4_	7.74	4.28
10	[Cu_0.25_(Al, Fe, Co, Cr, Ni)_0.75_]_3_O_4_	5.34	3.58
11	[Mn_0.35_(Al, Fe, Co, Cr, Ni)_0.65_]_3_O_4_	3.82	3.84
12	[Mn_0.25_(Al, Fe, Co, Cr, Ni)_0.75_]_3_O_4_	2.65	3.12

**Table 3 molecules-27-05951-t003:** The geometrically normalized peak current values for WOR using all the HEOs at 100 mV s^−1^ in 1M KOH + 1M CH_3_OH.

HEOs	Jpa (mA cm^−2^)	HEOs	Jpa (mA cm^−2^)	HEOs	Jpa (mA cm^−2^)
Ak-1	61.7	AS-1	19.2	MMCu-35	62.2
Ak-F	68.0	AS-F	58.4	MMCu-25	58.9
Ak-A	66.8	AS-A	68.0	MMMn-35	49.8
Ak-M	52.3	AS-M	52.3	MMMn-25	48.6

## Data Availability

Not applicable.

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
