# Peer review of "Sol-Gel Synthesized High Entropy Metal Oxides as High-Performance Catalysts for Electrochemical Water Oxidation"

_molecules, 2022, doi:10.3390/molecules27185951_

Round 1
Reviewer 1 Report
This paper elucidates the applicability of high entropy metal oxides as electrocatalysts for electrochemical water oxidation. It’s well written and comprehends the voltammetric output in a systematic manner. I would like to recommend this paper for publication after some minor comments.
1. The numbers given to each author aren’t in a correct way. All the authors belonging to institute 1 should be superscript as 1 and the others in same way.
2. (NEHEOs) HEOs in line 31 (Abstract) should be (NEHEOs) and also simplify this statement.
3. Indicate the major numerical output in abstract to make it more profound.
4. Cite recently reported synthesis approaches for HEOs in introduction part.
5. Put section 3 of materials and methods before results and discussion.
6. In table 3, there are some new representations (Ak-1, MMCu etc). Cross check it and indicate the accurate naming for the HEOs.
This paper elucidates the applicability of high entropy metal oxides as electrocatalysts for electrochemical water oxidation. It’s well written and comprehends the voltammetric output in a systematic manner. I recommend the paper for publication after some comments are addressed.
1. The numbers given to each author aren’t in a correct way. All the authors belonging to institute 1 should be superscript as 1 and the others in same way.
2. (NEHEOs) HEOs in line 31 (Abstract) should be (NEHEOs) and also simplify this statement.
3. Indicate the major numerical output in abstract to make it more profound.
4. Cite recently reported synthesis approaches for HEOs in introduction part.
5. Put section 3 of materials and methods before results and discussion.
6. In table 3, there are some new representations (Ak-1, MMCu etc). Cross check it and indicate the accurate naming for the HEOs.
Author Response
Open Review 1.
Comments and Suggestions for Authors
This paper elucidates the applicability of high entropy metal oxides as electrocatalysts for electrochemical water oxidation. It’s well written and comprehends the voltammetric output in a systematic manner. I would like to recommend this paper for publication after some minor comments.
Authors are thankful to worthy reviewers for the review and suggestions. The suggestions have been incorporated in the manuscript and replied to the queries is produced here:
- The numbers given to each author aren’t in a correct way. All the authors belonging to institute 1 should be superscript as 1and the others in same way.
Reply: The institute numbering has been corrected now.
- (NEHEOs) HEOs in line 31 (Abstract) should be (NEHEOs) and also simplify this statement.
Reply: The correction has been done and suggested editing has also been carried out.
- Indicate the major numerical output in abstract to make it more profound.
Reply: The major numerical output has been added in the abstract and edited, as well.
- Cite recently reported synthesis approaches for HEOs in introduction part.
Reply: Recently reported synthesis approaches for HEOs have been replaced/added in the manuscript.
- Put section 3 of materials and methods before results and discussion.
Reply: Section 3 of materials and methods has been shifted as suggested by the worthy reviewers now as Section 2 (although the templated format was used earlier).
- In table 3, there are some new representations (Ak-1, MMCu etc). Cross check it and indicate the accurate naming for the HEOs.
Reply: The codes have been specified in Table 1.
Author Response
This work reports Sol-Gel Synthesized High Entropy Metal Oxides as High-Performance Catalysts for Electrochemical Water Oxidation. This is interesting investigation with good results. However, following few concerns should be addressed before considering the manuscript for publication.
Reply: Author are grateful for the review and editing suggestions. The manuscript has been checked/edited throughout and the grammatical typos/errors were rectified. The reply/rebuttal is produced in the manuscript and here.
- The sol-gel method is not eco-friendly and not cost-effective. So, what is the commercialization point of this method?
Reply: In general, most of the synthesis methods are associated with benefits and drawbacks with environmental effects. Sol-gel method is an ample and proven scheme to get nearly homogeneous and uniformly distributed nanoparticles. The particle size of 30 nm was achieved via solgel method in comparison to 50 nm sized particles via ball-milling [Thuy, N.T. and D.L. Minh, Size effect on the structural and magnetic properties of nanosized perovskite LaFeO3 prepared by different methods. Advances in Materials Science and Engineering, 2012. 2012(380306): pages 6.]. The solution combustion synthesis (SCS) or auto-combustion is a cotemporary method to sol-gel and requires more experimental intricacies.
Reference “Chen, H., et al., Entropy stabilized metal oxide solid solutions as CO oxidation catalysts with high-temperature stability. Journal of Materials Chemistry A, (2018) 6: p. 11129-11133” has reported that the particle sizes increase, and agglomeration is a consequence of the high temperature sintering above 850 °C. The solid-state synthesis schemes involve high temperature sintering step.
Therefore, sol-gel method has more advantages than the solid-state reaction or arc sintering process.
- Why did authors used KCl solution for the electrochemical studies of the as-prepared electrode?
Reply: KCl is a neutral electrolyte and act as supporting medium to minimize the migration current effects. It is used in combination with the model analyte, potassium hexacyanoferrate, (K4[Fe(CN)6]) to estimate the active area of the modified electrodes (not used in this scheme).
How long the as-synthesized electrode can be used and is it recyclable? Authors are suggested to provide stability/durability of the as-synthesized electrode?
Reply: Section 3.4.7 Stability Test has been added with 3600 s durability. The stability test for the fabricated electrodes were conducted in 1 M KOH applying 1.4 V potential for 3600 s in chronoamperometric way. The catalytic platforms were found stable over the experimental time scale.
- The authors have done the oxidation process of these materials in KOH solution. As KOH is a toxic and nonproductive material, why does the author use it? Secondly, is the KI cheaper than KOH? Compare the analysis for them.
Reply: KOH provides a basic medium that is necessary for water oxidation process to provide electrochemically (EC) pure ingredients (e.g., O2) for the fuel cells utility. Although, the EC water splitting has been researched in our labs for many years now utilizing various media or electrolyte, the basic medium requirement is evident in the precise WOR or OER peak observation. KOH is like a universal medium for the OER/WOR and serves as OH- ions source under ambient conditions. The optimal KOH concentration has been used and the other media like KI isn’t helpful in the water electrolysis. Ref 54 also details some of the mechanistic involvement of the OH- ions in the WOR process. In future, various neutral electrolytes will be explored and used for such research.
Some minor comments:
- There are numerous typos and grammatical errors. Authors are suggested to revise the manuscript to improve the readability.
Reply: The manuscript has been checked/edited throughout and the grammatical typos/errors were rectified.
- The introduction lacks sufficient references. Authors are suggested to improve the citations with recent literature. Some suggested references. ChemElectroChem 4 (12), 3126-3133, Science Bulletin 66 (21), 2207-2216, Chinese Journal of Catalysis 43 (6), 1459-1472, Angew. Chem. Int. Ed. 2022, 61, 202115835-202115843.
Reply: A few more references have been replaced/added in discussion or other parts of the manuscript. See green highlighted numbers in the references list.
